# ACTIVE CONTINUAL LEARNING: ON BALANCING KNOWLEDGE RETENTION AND LEARNABILITY

## ABSTRACT

Acquiring new knowledge without forgetting what has been learned in a sequence of tasks is the central focus of continual learning (CL). While tasks arrive sequentially, the training data are often prepared and annotated independently, leading to the CL of incoming supervised learning tasks. This paper considers the under-explored problem of active continual learning (ACL) for a sequence of active learning (AL) tasks, where each incoming task includes a pool of unlabelled data and an annotation budget. We investigate the effectiveness and interplay between several AL and CL algorithms in the domain, class and task-incremental scenarios. Our experiments reveal the trade-off between two contrasting goals of not forgetting the old knowledge and the ability to quickly learn new knowledge in CL and AL, respectively. While conditioning the AL query strategy on the annotations collected for the previous tasks leads to improved task performance on the domain and task incremental learning, our proposed forgetting-learning profile suggests a gap in balancing the effect of AL and CL for the class-incremental scenario.

## 1 INTRODUCTION

The ability to continuously acquire knowledge while retaining previously learned knowledge is the hallmark of human intelligence. The pursuit to achieve this type of learning is referred to as continual learning (CL). Standard CL protocol involves the learning of a sequence of incoming tasks where the learner has limited access to training data from the previous tasks, posing the risk of forgetting past knowledge. Despite the sequential learning nature in which the learning of previous tasks may heavily affect the learning of subsequent tasks, the standard protocol often ignores the process of training data collection. That is, it implicitly assumes independent data annotation among tasks without considering the learning dynamic of the current model.

In this paper, we explore active learning (AL) problem to annotate training data for CL, namely active continual learning (ACL). AL and CL emphasise two distinct learning objectives (Mundt et al., 2020). While CL aims to maintain the learned information, AL concentrates on identifying suitable labelled data to incrementally learn new knowledge. The challenge is how to balance the ability of learning new knowledge and the prevention of forgetting the old knowledge (Riemer et al., 2019). Despite of this challenge, current CL approaches mostly focus on overcoming catastrophic forgetting — a phenomenon of sudden performance drop in previously learned tasks during learning the current task (McCloskey & Cohen, 1989; Ratcliff, 1990; Kemker et al., 2018).

Similar to CL, ACL also faces a similar challenge of balancing the prevention of catastrophic forgetting and the ability to *quickly* learn new tasks. Thanks to the ability to prepare its own training data proactively, ACL opens a new opportunity to address this challenge by selecting samples to both improve the learning of the current task and minimise interference to previous tasks. This paper conducts an extensive analysis to study the ability of ACL with the combination of existing AL and CL methods to address this challenge. We first investigate the benefit of actively labelling training data on CL and whether conditioning labelling queries on previous tasks can accelerate the learning process. We then examine the influence of ACL on balancing between preventing catastrophic forgetting and learning new knowledge.

Out contributions and findings are as follows:

- We formalise the problem of active continual learning and study the combination of several and prominent active learning and continual learning algorithms on image and text classification tasks covering three continual learning scenarios: domain, class and task incremental learning.

- We found that ACL methods that utilise AL to carefully select and annotate only a portion of training data can reach the performance of CL on the full training dataset, especially in the domain-IL (incremental learning) scenario.

- We observe that there is a trade-off between forgetting the knowledge of the old tasks and quickly learning the knowledge of the new incoming task in ACL. We propose the *forgetting-learning profile* to better understand the behaviour of ACL methods and discover that most of them are grouped into two distinct regions of *slow learners with high forgetting rates* and *quick learners with low forgetting rates*.

- ACL with sequential labelling is more effective than independent labelling in the domain-IL scenario where the learner can benefit from positive transfer across domains. In contrast, sequential labelling tends to focus on accelerating the learning of the current task, resulting in higher forgetting and lower overall performance in the class and task-IL.

- Our study suggests guidelines for choosing AL and CL algorithms. Across three CL scenarios, experience replay (Rolnick et al., 2019) is consistently the best overall CL method. Uncertainty-based AL methods perform best in the domain-IL scenario while diversity-based AL is more suitable for class-IL due to the ill-calibration of model prediction on newly introduced classes.

## 2 KNOWLEDGE RETENTION AND QUICK LEARNABILITY

This section first provides the problem formulation of continual learning (CL), active learning (AL) and active continual learning (ACL). We then present the evaluation metrics to measure the level of forgetting the old knowledge, the ability to quickly learn new knowledge and overall task performance.

### 2.1 KNOWLEDGE RETENTION IN CONTINUAL LEARNING

**Continual Learning** It is the problem of learning a sequence of tasks $\mathcal{T} = \{\tau_1, \tau_2, \cdots, \tau_T\}$ where $T$ is the number of tasks with an underlying model $f_\theta(.)$. Each task $\tau_t$ has training data $\mathcal{D}_t^l = \{(x_i^t, y_i^t)\}$ where $x_i^t$ is the input drawn from an input space $\mathcal{X}_t$ and its associated label $y_i^t$ in label space $\mathcal{Y}_t$, sampled from the joint input-output distribution $P_t(\mathcal{X}_t, \mathcal{Y}_t)$. At each CL step $1 \leq t \leq T$, task $\tau_t$ with training data $\mathcal{D}_t^l$ arrives for learning while only a limited subset of training data $\mathcal{D}_{t-1}^l$ from the previous task $\tau_{t-1}$ are retained. We denote $\theta_t^*$ as the model parameter after learning the current task $\tau_t$ which is continually optimised from the parameter $\theta_{t-1}^*$ learned for the previous tasks, $f_{\theta_t^*} = \psi(f_{\theta_{t-1}^*}, \mathcal{D}_t^l)$ where $\psi(.)$ is the CL algorithm. The objective is to learn the model $f_{\theta_t^*}$ such that it achieves good performance for not only the current task $\tau_t$ but also all previous tasks $\tau_{<t}$, $\frac{1}{t} \sum_{i=1}^t A(\mathcal{D}_i^{test}, f_{\theta_t^*})$ where $\mathcal{D}_t^{test}$ denotes the test set of task $\tau_t$, and $A(.)$ is the task performance metric (e.g. the accuracy).

Depending on the nature of the tasks, CL problems can be categorised into domain, class and task incremental learning.

- **Domain incremental learning (domain-IL)** refers to the scenario where all tasks in the task sequence differ in the input distribution $P_{t-1}(\mathcal{X}_{t-1}) \neq P_t(\mathcal{X}_t)$ but share the same label set $\mathcal{Y}_{t-1} = \mathcal{Y}_t$.

- **Class incremental learning (class-IL)** is the scenario where new classes are added to the incoming task $\mathcal{Y}_{t-1} \neq \mathcal{Y}_t$. The model learns to distinguish among classes not only in the current task but also across previous tasks.

- **Task incremental learning (task-IL)** assists the model in learning a sequence of non-overlapping classification tasks. Each task is assigned with a unique id which is then added to its data samples so that the task-specific parameters can be activated accordingly.

**Forgetting Rate** We denote $A_{i,j}$ as the performance of task $\tau_j$ after training on the tasks up to $\tau_i$, i.e. $A_{i,j} := A(\mathcal{D}_j^{test}, f_{\theta_i^*})$. The overall task performance is measured by the average accuracy at the

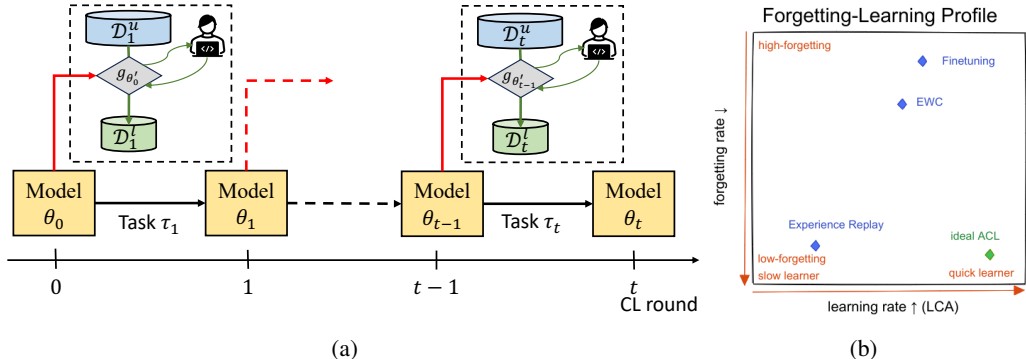

Figure 1: (a) Active continual learning (ACL) annotates training data sequentially by conditioning on the learning dynamic of the current model (red arrow). (b) Forgetting-Learning Profile to visualize the balance between old knowledge retention and new knowledge learning in ACL. An ideal ACL method should lie at the quick learner with low forgetting rate region.

end of CL procedure $\frac{1}{T}\sum_{j=1}^{T} A_{T,j}$ where $T$ is the number of tasks. Following the literature, we use forgetting rate (Chaudhry et al., 2018) as the forgetting measure. It is the average of the maximum performance degradation due to learning the current task over all tasks in the task sequence

$$FR = \frac{1}{T-1} \sum_{j=1}^{T-1} \max_{k \in [j,T]} A_{k,j} - A_{T,j} \qquad (1)$$

The lower the forgetting rate, the better the ability of the model to retain knowledge.

## 2.2 QUICK LEARNABLITY IN ACTIVE LEARNING

The core research question in pool-based AL is how to identify informative unlabelled data to annotate within a limited annotation budget such that the performance of an underlying active learner is maximised. Given a task $\tau$ with input space $\mathcal{X}$ and label space $\mathcal{Y}$, the AL problem often starts with a small set of labelled data $\mathcal{D}^l = \{(x_i, y_i)\}$ and a large set of unlabelled data $\mathcal{D}^u = \{x_j\}$ where $x_i, x_j \in \mathcal{X}, y_i \in \mathcal{Y}$. The AL procedure consists of multiple rounds of selecting unlabelled instances to annotate and repeats until exhausting the annotation budget.

More specifically, the algorithm chooses one or more  instances $x^*$ in the unlabelled dataset $\mathcal{D}^u$ to ask for labels in each round according to an scoring/acquisition function $g(.)$, $x^* = \arg\max_{x_j \in \mathcal{D}^u} g(x_j, f_\theta)$ where the acquistion function estimates the value of an unlabeled data point, if labelled, to the re-training of the current model. The higher the score of an unlabelled datapoint, the higher performance gain it may bring to the model. The differences between AL algorithms boil downs to the choice of the scoring function $g(.)$.

**Quick Learnability** Learning curve area (LCA) (Chaudhry et al., 2019a) is the area under the accuracy curve of the current task with respect to the number of trained minibatches. It measures how quickly a learner learns the current task. The higher the LCA value, the quicker the learning is. We adopt this metric to compute the learning speed of an ACL method wrt the amount of annotated data.

## 2.3 ACTIVE CONTINUAL LEARNING

We consider the problem of continually learning a sequence of AL tasks, namely active continual learning (ACL). Figure 1a illustrates the learning procedure of the ACL problem. CL often overlooks how training data is annotated and implicitly assumes independent labelling among tasks, leading to CL as a sequence of *supervised learning* tasks.

More specifically, each task $\tau_t$ in the task sequence consists of an initial labelled dataset $\mathcal{D}_t^l$, a pool of unlabelled data $\mathcal{D}_t^u$ and an annotation budget $B_t$. The ACL training procedure is described

---

**Algorithm 1** Active Continual Learning

---

**Input:** Task sequence $\mathcal{T} = \{\tau_1, \cdots, \tau_T\}$ where $\tau_t = \{\mathcal{D}_t^l, \mathcal{D}_t^u, \mathcal{D}_t^{test}, B_t\}$, initial model $\theta_0$, query size $b$
**Output:** Model parameters $\theta$

1: Initialize $\theta_0^*$
2: **for** $t \in 1, \ldots, T$ **do** $\qquad\qquad\qquad\qquad\qquad\qquad\qquad\qquad\qquad\qquad\qquad$ ▷ *CL loop*
3: $\quad$ $f_{\hat{\theta}_t} \leftarrow \psi(\mathcal{D}_t^l, f_{\theta_{t-1}^*})$ $\qquad\qquad\qquad$ ▷ *Build the proxy model to be used in the AL acquisition function*
4: $\quad$ **for** $i \in 1, \ldots, \frac{B_t}{b}$ **do** $\qquad\qquad\qquad\qquad\qquad\qquad\qquad\qquad\qquad\qquad$ ▷ *AL round*
5: $\qquad$ $\{x_j^*\}|_1^b \leftarrow g(\mathcal{D}_t^l, \mathcal{D}_t^u, b, f_{\hat{\theta}_t})$ $\qquad\qquad\qquad\qquad\qquad\qquad$ ▷ *Build AL query*
6: $\qquad$ $\{(x_j^*, y_j^*)\}|_1^b \leftarrow \text{annotateByOracle}(\{x_j\}|_1^b)$
7: $\qquad$ $\mathcal{D}_t^u \leftarrow \mathcal{D}_t^u \backslash \{x_j^*\}|_1^b$ $\qquad\qquad\qquad\qquad\qquad\qquad\qquad$ ▷ *Update unlabelled dataset*
8: $\qquad$ $\mathcal{D}_t^l \leftarrow \mathcal{D}_t^l \cup \{(x_j^*, y_j^*)\}|_1^b$ $\qquad\qquad\qquad\qquad\qquad\qquad$ ▷ *Update labelled dataset*
9: $\qquad$ $f_{\hat{\theta}_t} \leftarrow \psi(\mathcal{D}_t^l, f_{\theta_{t-1}^*})$ $\qquad\qquad\qquad\qquad\qquad\qquad\qquad$ ▷ *Re-train the proxy model*
10: $\quad$ **end for**
11: $\quad$ $f_{\theta_t^*} \leftarrow \psi(\mathcal{D}_t^l, f_{\theta_{t-1}^*})$ $\qquad$ ▷ *Re-train the model on the AL collected data starting from the previous task*
12: **end for**
13: **return** $\theta_T^*$

---

in Algorithm 1. Upon the arrival of the current task $\tau_t$, we first train the *proxy model* $f_{\hat{\theta}_t}$ on the current labelled data with the CL algorithm $\psi(.)$ from the best checkpoint of previous task $\theta_{t-1}^*$ (line 3). This proxy model is going to be used later in the AL acquisition function. We then iteratively run the AL query to annotate new labelled data (lines 4-8) and retrain the proxy model (line 9) until exhausting the annotation budget. The above procedure is repeated for all tasks in the sequence. Notably, the AL acquisition function $g(.)$ ranks the unlabelled data (line 5) according to the proxy model $f_{\hat{\theta}_t}$, which is warm-started from the model learned from the previous tasks. Hence, the labelling query in ACL is no longer independent from the previous CL tasks and is able to leverage their knowledge.

Another active learning approach in continual learning would be to have an acquisition function for each task, which does not depend on the previous tasks. That is to build the proxy model (lines 3 and 9) by iniliazing it *randomly*, instead of the model learned from the previous tasks. We will compare this *independent* AL approach to the ACL approach in the experiments, and show that leveraging the knowledge of the previous tasks is indeed beneficial for some CL settings.

**Forgetting-Learning Profile** To understand the trade-off between CL and AL, we propose the forgetting-learning profile - a 2-D plot where the x-axis is the LCA, the y-axis is the forgetting rate, and each ACL algorithm represents a point in this space (Figure 1b). Depending on the level of forgetting and the learning speed, the forgetting-learning space can be divided into four regions: slow learners with low/high forgetting rate which corresponding the bottom and top left quarters; and similarly, quick learners with low/high forgetting rate residing at the bottom and top right quarters. Ideally, an effective ACL should lie at the quick learner with low forgetting rate region.

## 3 EXPERIMENTS

In this paper, we investigate the effectiveness and dynamics of ACL by addressing the following research questions (**RQs**):

- **RQ1**: Does utilising AL to carefully select and annotate training data improve CL performance?

- **RQ2**: Is it more effective to label data sequentially than CL with independent AL?

- **RQ3**: How do different ACL methods influence the balance between forgetting and learning?

**Dataset** We conduct ACL experiments on two text classification and three image classification tasks. The text classification tasks include aspect sentiment classification (ASC) (Ke et al., 2021) and news classification (20News) (Pontiki et al., 2014), corresponding to the domain and class/task incremental learning, respectively. For image classification tasks, we evaluate the domain-IL scenario with the permuted-MNIST (P-MNIST) dataset and class/task IL scenaroios with the sequential

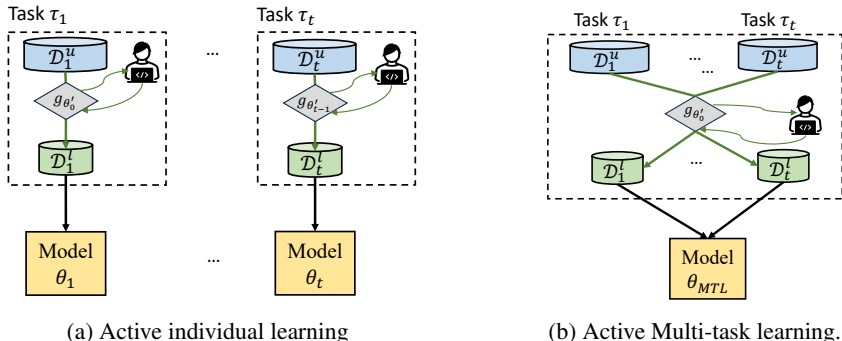

(a) Active individual learning        (b) Active Multi-task learning.

Figure 2: The ceiling methods of ACL.

MNIST (S-MNIST) (Lecun et al., 1998) and sequential CIFAR10 (S-CIFAR10) datasets (Krizhevsky et al., 2009). The detail of task grouping and data statistics are reported in Table 3 in Appendix C.

**Continual Learning Methods** This paper mainly studies two widely-adopted methods in CL literature: elastic weight consolidation (EWC) (Kirkpatrick et al., 2017) and experience replay (ER) (Rolnick et al., 2019). EWC adds a regularization term based on the Fisher information to constrain the update of important parameters for the previous tasks. ER exploits a fixed replay buffer to store and replay examples from the previous tasks during training the current task. We also evaluate other experience replay CL methods including iCaRL (Rebuffi et al., 2017), AGEM (Chaudhry et al., 2019a), GDumb (Prabhu et al., 2020a), DER and DER++ (Buzzega et al., 2020) on image classification benchmarks. The detailed description of each CL method is reported in Appendix A.

**Active Learning Methods** We consider two uncertainty-based methods, including entropy (ENT) and min-margin (MARG) (Scheffer et al., 2001; Luo et al., 2005), embedding-$k$-means (KMEANS) - a diversity-based AL method (Yuan et al., 2020), coreset (Sener & Savarese, 2018) and BADGE which takes both uncertainty and diversity into account (Ash et al., 2020), and random sampling as AL baseline. While the uncertainty-based strategy gives a higher score to unlabelled data deemed uncertain to the current learner, the diversity-based method aims to increase the diversity among selected unlabelled data to maximise the coverage of representation space. The detailed description of each AL method is reported in Appendix B.

**Ceiling Methods** We report the results of AL in building different classifiers for each individual task (Figure 2.a). This serves as an approximate ceiling method for our ACL setting. We also report the integration of AL methods with multi-task learning (MTL), as another approximate ceiling method (Figure 2.b). In each AL round, we use the current MTL model in the AL acquisition function to estimate the scores of unlabelled data for *all* tasks *simultaneously*. We then query the label of those unlabelled data points which are ranked high and the labelling budget of their corresponding tasks are not exhausted. These two ceiling methods are denoted by MTL (multitask learning) and INDIV (individual learning) in the result tables.

**Model Training and Hyperparameters** We experiment cold-start AL for AL and ACL models. That is, the models start with an empty labelled set and select 1% training data for annotation until exhausting the annotation budget, i.e. 30% for ASC, 20% for 20News, 25% for S-CIFAR10 dataset. For P-MNIST and S-MNIST, the query size and the annotation budget are 0.5% and 10%. While the text classifier is initialised with RoBERTa (Liu et al., 2019), image classifiers (MLP for P-MNIST and S-MNIST, Resnet18 (He et al., 2016) for S-CIFAR10) are initialized randomly. During training each task, the validation set of the corresponding task is used for early stopping. We report the average results over 6 runs with different seeds for the individual learning and multi-task learning baselines. As CL results are highly sensitive to the task order, we evaluate each CL and ACL method on 6 different task orders and report the average results. More details on training are in Appendix D.

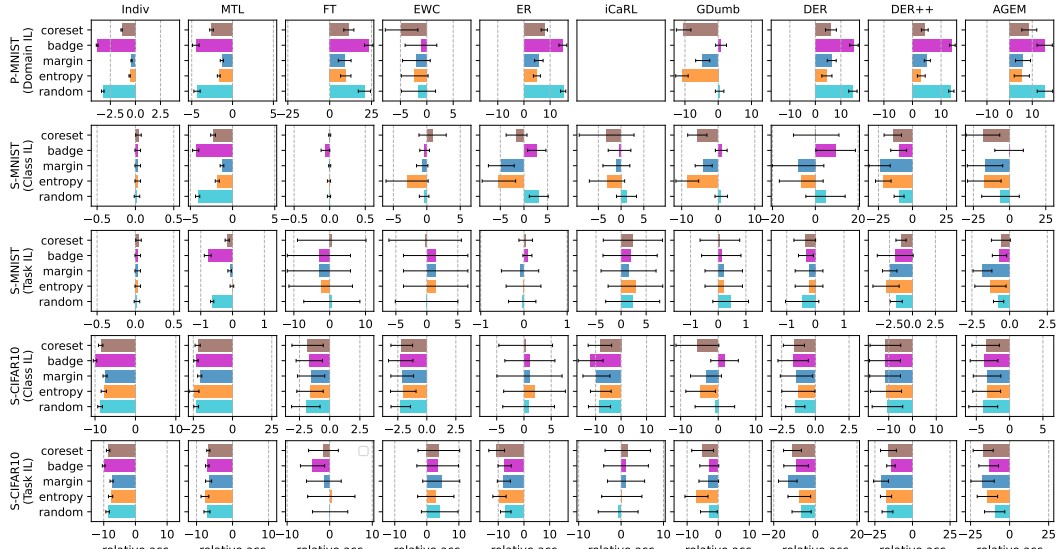

Figure 3: Relative performance (average accuracy of 6 runs) of various ACL methods with respect to the CL on full labelled data (full CL) in image classification benchmarks. The error bar indicates the standard deviation of the difference between two means, full CL and ACL. iCaRL is a class-IL method, hence not applicable for P-MNIST dataset.

## 3.1 ACTIVE CONTINUAL LEARNING RESULTS

**Image Classification**  The relative average task performance of ACL methods with respect to CL on full labelled data in three image classification benchmarks are reported in Figure 3. Detailed results are shown in Table 6 in Appendix E. Notably, in the P-MNIST dataset (domain-IL scenario), only the CL ceiling methods (Indv and MTL) on the full dataset outperform their active learning counterparts. While utilising only a portion of training data (10-30%), ACL methods surpass the performance of the corresponding CL methods in domain-IL scenario and achieve comparable performance in most class/task-IL scenarios (with only 2-5% difference), except in S-CIFAR10 Task-IL. As having access to a small amount of training data from previous tasks, experience replay methods are less prone to catastrophic forgetting and achieve higher accuracy than naive finetuning (FT) and the regularization method EWC. Within the same CL method, uncertainty-based AL methods (ENT and MARG) generally perform worse than other AL methods.

**Text Classification**  Figure 4 shows the relative performance with respect to CL on full labelled data of text classification benchmarks. Detailed accuracy is reported in Table 4 in Appendix E. In ASC dataset, the multi-task learning, CL and ACL models outperform individual learning (INDIV), suggesting positive transfer across domains. In contrast to the observations in P-MNIST, uncertainty-based AL methods demonstrate better performance than other AL methods in the domain-IL scenario. Conversely, there is no significant difference in accuracy when comparing various AL methods in the 20News - class IL scenario. On the other hand, ACL significantly lags behind supervised CL in 20News task-IL scenario, especially in EWC. Detailed accuracy and standard deviation are provided in Appendix E.

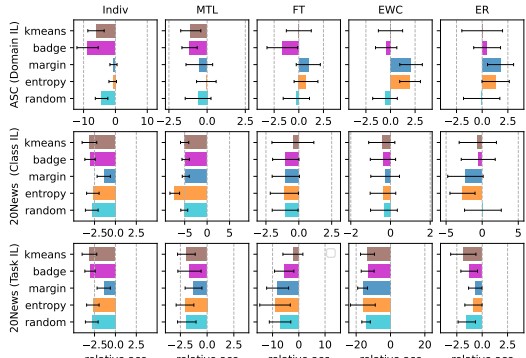

Figure 4: Relative performance (average accuracy of 6 runs) of various ACL methods with respect to the CL on full labelled data in text classification tasks. The error bar indicates the standard deviation of the difference between two means, full CL and ACL.

Table 1: Relative performance of CL with independent AL with respect to the corresponding ACL. DIVER denotes diversity-based method, kMeans for text classification and coreset for image classification tasks.

| | ASC (Domain-IL) | | | 20News (Class-IL) | | | 20News (Task-IL) | | | S-MNIST (Class-IL) | | | S-MNIST (Task-IL) | | |
| | FT | EWC | ER | FT | EWC | ER | FT | EWC | ER | FT | EWC | ER | FT | EWC | ER |
|---|---|---|---|---|---|---|---|---|---|---|---|---|---|---|---|
| IND. RAND | −1.67 | +0.35 | −0.02 | −0.03 | +0.01 | +0.90 | +1.70 | −3.22 | +0.23 | −0.10 | +3.60 | +0.70 | +3.82 | −1.33 | −0.12 |
| IND. ENT | −0.59 | −0.92 | −0.19 | −0.09 | −0.08 | +2.32 | −0.87 | +3.39 | −1.03 | −0.30 | +3.59 | +1.83 | −0.35 | −10.70 | −0.63 |
| IND. MARG | −0.93 | −1.36 | −0.82 | −0.08 | +0.14 | +2.24 | +1.05 | +2.58 | −0.42 | −0.19 | −1.94 | +0.06 | +1.68 | −11.91 | −0.52 |
| IND. BADGE | +0.15 | −0.08 | +0.10 | +0.01 | −0.20 | +4.28 | −1.16 | +3.14 | −0.38 | −0.06 | +6.14 | +0.59 | +3.35 | −0.64 | −0.47 |
| IND. DIVER | −0.54 | −0.71 | +0.96 | −0.21 | +0.06 | +2.78 | −2.66 | +3.65 | +0.35 | −0.07 | +0.66 | −0.60 | −6.58 | −4.94 | −0.57 |

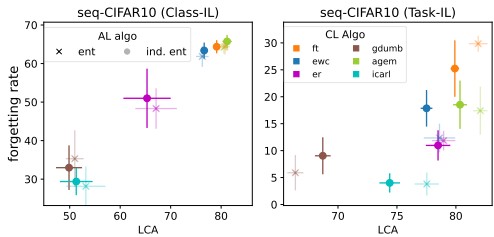

Figure 5: Learning-Forgetting profile of ACL methods with entropy in text classification tasks.

**Comparison to CL with Independent AL**    We have shown the benefit of active continual learning over the supervised CL on full dataset in domain and class-IL for both image and text classification tasks. An inherent question that arises is whether to actively annotate data for each task independently or to base it on the knowledge acquired from previous tasks. Table 1 shows the difference of the performance between CL with independent AL and the corresponding ACL method, i.e. negative values show ACL is superior. Interestingly, the effect of sequential labelling in ACL varies depending on the CL scenarios. For the ASC dataset, the performance of ACL is mostly better than CL with independent AL, especially for uncertainty-based AL methods (ENT and MARG). In contrast, we do not observe the improvement of ACL over the independent AL for the 20News and S-MNIST dataset in the class-IL scenario. On the other hand, sequential AL seems beneficial in experience replay methods in task-IL scenarios. Full results in other CL methods are provided in Appendix E.3.

## 3.2 KNOWLEDGE RETENTION AND QUICK LEARNABILITY

Figure 5 and Figure 6 report the forgetting rate and LCA for ACL methods with entropy strategy for text classification tasks and S-CIFAR10. The results of other ACL methods can be found in **??**. These metrics measure the ability to retain knowledge and the learning speed on the current task. Ideally, we want to have a quick learner with the lowest forgetting. ER in class-IL scenario has lower LCA than FT and EWC due to the negative interference of learning a minibatch of training data from both the current and previous tasks. However, it has much lower

Figure 6: Learning-Forgetting profile of ACL methods with entropy on seq-CIFAR10 dataset.

forgetting rate. In domain-IL, ER has a comparable forgetting rate and LCA with other ACL methods where positive domain transfer exists. On labelling strategy, sequential labelling results in quicker learners than independent labelling across all three learning scenarios. This evidence shows the benefit of carefully labelling for the current task by conditioning on the previous tasks. However, it comes with a compensation of a slightly higher forgetting rate.

**Learning Rate**    Figure 7 show the average LCA of AL curves for entropy-based ACL methods. For each task $\tau_t$, we compute the LCA of the average accuracy curve of all learned tasks $\tau_{\leq t}$ at each AL round. This metric reflects the learning speed of the models on all seen tasks so far. The higher the LCA, the quicker the model learns. In domain-IL, ACL with sequential labelling has a higher LCA than independent labelling across 3 CL methods. The gap is larger for early tasks and shrunk towards the final tasks. Unlike the increasing trend in domain-IL, we observe a decline in LCA over tasks in all ACL methods in class and task-IL. This declining trend is evidence of catastrophic forgetting.

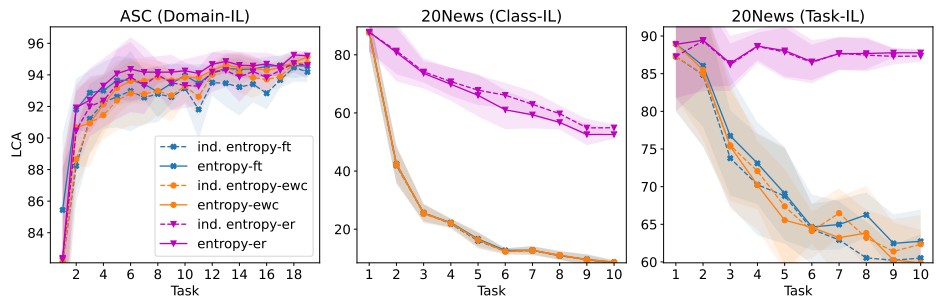

Figure 7: Average LCA of AL curves through tasks.

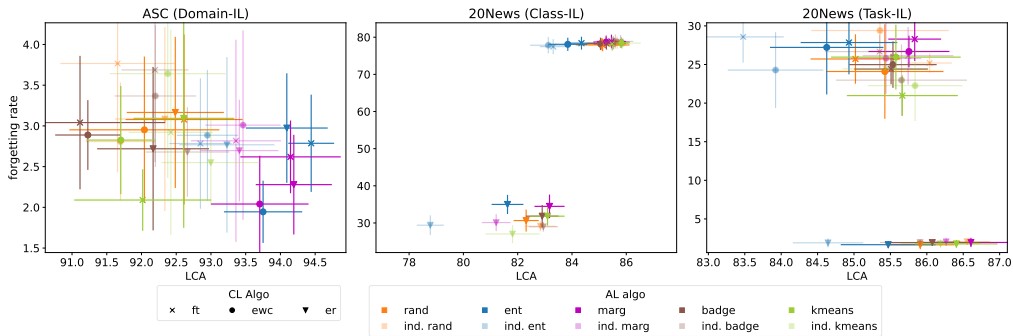

Figure 8: Learning-Forgetting profile of ACL methods for text classification tasks.

Aligned with above findings that sequential labelling suffers from more severe forgetting, ER with independent labelling has better LCA than sequential labelling, especially at the later tasks.

**Forgetting-Learning Profile**  Figure 8 and Figure 9 show the forgetting-learning profile for ACL models trained on image and text classification benchmarks respectively. In text classification tasks (Figure 8), while the ACL methods scatter all over the space in the profile of domain-IL for ASC dataset, the sequential labelling with uncertainty (ENT and MARG) and ER, EWC has the desired properties of low forgetting and quick learning. For task-IL, ER also lies in the low forgetting region, especially several ER with AL methods also have quick learning ability. On the other hand, we can observe two distinct regions in the profile of class-IL: the top right region of quick learners with high forgetting (FT and EWC), and the bottom left region of slow learners with low forgetting (ER). While ACL with ER has been shown effective in non-forgetting and quick learning ability for domain and task-IL, none of the studied ACL methods lies in the ideal regions in the class-IL profile. Compared to sequential labelling, independent labelling within the same ACL methods tend to reside in slower learning regions but with lower forgetting. We observe similar findings in the S-CIFAR10 dataset (Figure 9). However, in the case of S-MNIST, most experience replay methods reside in the ideal low-forgetting and quick learning region. We hypothesize that this phenomenon can be attributed to the relatively simpler nature of MNIST datasets, where most methods achieve nearly perfect accuracy.

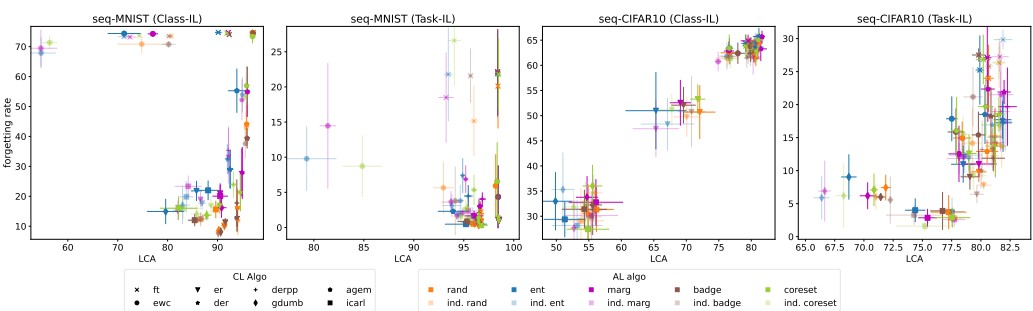

Figure 9: Learning-Forgetting profile of ACL methods for image classification tasks.

**Normalized forgetting rate at different AL budget** We have shown that there is a trade-off between forgetting and quick-learnability when combining AL and CL. To test our hypothesis that sequential active learning with continual learning may lead to forgetting old knowledge, we report the normalized ratio of the forgetting rate of ACL over the forgetting rate of supervised CL $\frac{FR_{ACL}}{FR_{CL}}$ at different annotation budget.[1]

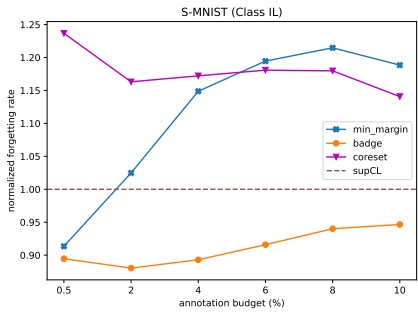

Figure 10 reports the average normalized forgetting ratio across different CL methods in S-MNIST. A normalized ratio greater than 1 means that the ACL method has higher forgetting than the corresponding supervised CL baseline, i.e. performing AL increases the forgetting. We observe that both diversity-based (coreset) and uncertainty-based (min-margin) methods lead to more forgetting than the baseline. On the other hand, BADGE consistently scores lower forgetting rates across different annotation budgets.

Figure 10: Normalized forgetting rate across different CL methods.

# 4 RELATED WORKS

**Active learning** AL algorithms can be classified into heuristic-based methods such as uncertainty sampling (Settles & Craven, 2008; Houlsby et al., 2011) and diversity sampling (Brinker, 2003; Joshi et al., 2009); and data-driven methods which learn the acquisition function from the data (Bachman et al., 2017; Fang et al., 2017; Liu et al., 2018; Vu et al., 2019). We refer readers to the following survey papers (Settles, 2012; Zhang et al., 2022) for more details.

**Continual learning** CL has rich literature on mitigating the issue of catastrophic forgetting and can be broadly categorised into regularisation-based methods (Kirkpatrick et al., 2017), memory-based method (Aljundi et al., 2019; Zeng et al., 2019; Chaudhry et al., 2019b; Prabhu et al., 2020b) and architectural-based methods(Hu et al., 2021; Ke et al., 2021). We refer the reader to the survey (Delange et al., 2021) for more details. Recent works have explored beyond CL on a sequence of supervised learning tasks such as continual few-shot learning (Li et al., 2022), unsupervised learning (Madaan et al., 2021). In this paper, we study the usage of AL to carefully prepare training data for CL where the goal is not only to prevent forgetting but also to quickly learn the current task.

# 5 CONCLUSION

This paper studies the under-explored problem of carefully annotating training data for continual learning (CL), namely active continual learning (ACL). Our experiments and analysis shed light on the performance characteristics and learning dynamics of the integration between several well-studied active learning (AL) and CL algorithms. With only a portion of the training set, ACL methods achieve comparable results with CL methods trained on the entire dataset; however, the effect on different CL scenarios varies. We also propose the forgetting-learning profile to understand the relationship between two contrasting goals of low-forgetting and quick-learning ability in CL and AL, respectively. We identify the current gap in ACL methods where the AL acquisition function concentrates too much on improving the current task and hinders the overall performance.

## LIMITATIONS

Due to limited computational resources, this study does not exhaustively cover all AL and CL methods proposed in the literature of AL and CL. We consider several well-established CL and AL algorithms and solely study on classification tasks. We hope that the simplicity of the classification tasks has helped in isolating the factors that may otherwise influence the performance and behaviours of ACL. We leave the exploration of learning dynamics of ACL with more complicated CL and AL algorithms and more challenging tasks such as sequence generation as future work.

---

[1]Normalized forgetting rates of different task orders in ACL are at the same scale, due to the normalization.

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

## A   CONTINUAL LEARNING METHODS

We consider the following continual learning baselines

- **Finetuning (FT)** is a naïve CL method which simply continue training the best checkpoint $\theta_{t-1}$ learned in previous task on the training data of the current task $\tau_t$.

- **Elastic Weight Consolidation (EWC)** (Kirkpatrick et al., 2017) is a regularisation method to prevent the model learned on task $\tau_{t-1}$ parametrised by $\theta_{t-1}$ from catastrophic forgetting when learning new task $\tau_t$. The overall loss for training on data $\mathcal{D}_t^l$ of task $\tau_t$ is

$$\mathcal{L}_\theta(\mathcal{D}_t) + \lambda \sum_i F_{i,i}(\theta_i - \theta_{t-1,i}^*) \tag{2}$$

where $\lambda$ is the coefficient parameter to control the contribution of regularisation term, $F_{i,i}(.)$ is the diagonal of the Fisher information matrix of the learned parameters on previous tasks $\theta_{t-1}^*$ and is approximate as the gradients calculated with data sampled from the previous task, that is

$$F_{i,i}(\theta) = \frac{1}{N} \sum_{x_j, y_j \in \mathcal{D}_{t-1}^l} \left( \frac{\partial \mathcal{L}(x_j, y_j; \theta)}{\partial \theta_i} \right)^2 \tag{3}$$

- **Experience Replay (ER)** (Rolnick et al., 2019) exploits a fixed replay buffer to store a small amount of training data in the previous tasks. Some replay examples are added to the current minibatch of current tasks for rehearsal during training

$$\mathcal{L}_{\theta_t}(\mathcal{D}_t) + \beta \mathbb{E}_{(x',y')\sim}[l(y', \hat{y}'; \theta_t)] \tag{4}$$

where $l(.)$ is the loss function and $\beta$ is a coefficient controlling the contribution of replay examples. Upon finishing the training of each task, some training instances from the current task are added to the memory buffer. If the memory is full, the newly added instances will replace the existing samples in the memory. In this paper, we maintain a fixed memory buffer of size $m = 400$ and allocate $\frac{m}{t}$ places for each task. We employ random sampling to select new training instances and for the sample replacement strategy.

- **DER** (Buzzega et al., 2020) and **DER++** (Buzzega et al., 2020) modify the experience replay loss with the logit matching loss for the samples from previous tasks.

- **iCaRL** (Rebuffi et al., 2017) is a learning algorithm for class-IL scenario. It learns class representation incrementally and select samples close to the class representation to add to the memory buffer.

- **GDumb**(Prabhu et al., 2020a) simply greedily train the models from scratch on only the samples from the memory buffer.

## B   ACTIVE LEARNING METHODS

We consider the following AL strategies in our experiment

- **Random sampling (RAND)** which selects query datapoints randomly.

- **Entropy (ENT)** strategy selects the sentences with the highest predictive entropy

$$f_{\text{AL}}(x) = -\sum \Pr_\theta(\hat{y}|x) \log \Pr_\theta(\hat{y}|x) \tag{5}$$

where $\hat{y} = \arg\max \Pr(y|x)$ is the predicted label of the given sentence $x$.

- **Min-margin (MARG)** strategy (Scheffer et al., 2001; Luo et al., 2005) selects the datapoint $x$ with the smallest different in prediction probability between the two most likely labels $\hat{y}_1$ and $\hat{y}_2$ as follows:

$$f_{\text{AL}}(x) = -(\Pr_\theta(\hat{y}_1|x) - \Pr_\theta(\hat{y}_2|x)) \tag{6}$$

- **BADGE** (Ash et al., 2020) measures uncertainty as the gradient embedding with respect to parameters in the output (pre-softmax) layer and then chooses an appropriately diverse subset by sampling via $k$-means++ (Arthur & Vassilvitskii, 2007).

| | Speaker | Router | Computer | Nokia6610 | Nikon4300 | Creative | CanonG3 | ApexAD | CanonD500 | Canon100 | Diaper | Hitachi | ipod | Linksys | MicroMP3 | Nokia6600 | Norton | Restaurant | Laptop |
|---|---|---|---|---|---|---|---|---|---|---|---|---|---|---|---|---|---|---|---|
| train | 352 | 245 | 283 | 271 | 162 | 677 | 228 | 343 | 118 | 175 | 191 | 212 | 153 | 176 | 484 | 362 | 194 | 2873 | 1730 |
| dev | 44 | 31 | 35 | 34 | 20 | 85 | 29 | 43 | 15 | 22 | 24 | 26 | 20 | 22 | 61 | 45 | 24 | 96 | 123 |
| test | 44 | 31 | 36 | 34 | 21 | 85 | 29 | 43 | 15 | 22 | 24 | 27 | 20 | 23 | 61 | 46 | 25 | 964 | 469 |

Table 2: Statistics of ASC dataset (domain incremental learning scenario).

| Task | 1 | | 2 | | 3 | | 4 | | 5 | | 6 | | 7 | | 8 | | 9 | | 10 | |
|---|---|---|---|---|---|---|---|---|---|---|---|---|---|---|---|---|---|---|---|---|
| | comp.graphics | comp.os.ms-windows.misc | comp.sys.ibm.pc.hardware | comp.sys.mac.hardware | comp.windows.x | rec.autos | rec.motorcycles | rec.sport.baseball | rec.sport.hockey | sci.crypt | sci.electronics | sci.med | sci.space | misc.forsale | talk.politics.misc | talk.politics.guns | talk.politics.mideast | talk.religion.misc | alt.atheism | soc.religion.christian |
| train | 800 | 800 | 800 | 800 | 800 | 800 | 800 | 800 | 800 | 800 | 800 | 800 | 800 | 800 | 800 | 800 | 800 | 800 | 800 | 797 |
| dev | 100 | 100 | 100 | 100 | 100 | 100 | 100 | 100 | 100 | 100 | 100 | 100 | 100 | 100 | 100 | 100 | 100 | 100 | 100 | 100 |
| test | 100 | 100 | 100 | 100 | 100 | 100 | 100 | 100 | 100 | 100 | 100 | 100 | 100 | 100 | 100 | 100 | 100 | 100 | 100 | 100 |

Table 3: Statistics of 20News dataset (class incremental learning scenario).

- **Embedding $k$-means (KMEANS)** is the generalisation of BERT-KM (Yuan et al., 2020) which uses the $k$-Mean algorithm to cluster the examples in unlabelled pool based on the contextualised embeddings of the sentences. The nearest neighbours to each cluster centroids are chosen for labelling. For the embedding $k$-means with MTL, we cluster the training sentences into $k \times T$ clusters and greedily choose a sentence from $k$ clusters for each task based on the distance to the centroids. In this paper, we compute the sentence embedding using hidden states of the last layer of RoBERTa (Liu et al., 2019) instead of BERT (Devlin et al., 2019).
- **coreset** (Sener & Savarese, 2018) is a diversity-based sampling method for computer vision task.

## C DATASET

**Text classification tasks**   ASC is a task of identifying the sentiment (negative, neutral, positive) of a given aspect in its context sentence. We use the ASC dataset released by (Ke et al., 2021). It contains reviews of 19 products collected from (Hu & Liu, 2004; Liu et al., 2015; Ding et al., 2008; Pontiki et al., 2014). We remove the neutral labels from two SemEval2014 tasks (Pontiki et al., 2014) to ensure the same label set across tasks. 20News dataset (Lang, 1995) contains 20 news topics, and the goal is to classify the topic of a given news document. We split the dataset into 10 tasks with 2 classes per task. Each task contains 1600 training and 200 validation and test sentences. The detail of task grouping of 20News dataset and the data statistics of both ASC and 20News datasets are reported in Appendix C. The data statistics of ASC and 20News dataset are reported in  Table 2 and Table 3, respectively.

**Image classification tasks**   The MNIST handwritten digits dataset (Lecun et al., 1998) contains 60K (approximately 6,000 per digit) normalised training images and 10K (approximately 1,000 per digit) testing images, all of size $28 \times 28$. In the sequential MNIST dataset (S-MNIST), we have 10 classes of sequential digits. In the sequential MNIST task, the MNIST dataset was divided into five tasks, such that each task contained two sequential digits and MNIST images were presented to the sequence model as a flattened $784 \times 1$ sequence for digit classification. The order of the digits was fixed for each task order. The CIFAR-10 dataset (Krizhevsky et al., 2009) contains 50K images for training and 10K for testing, all of size $32 \times 32$. In the sequential CIFAR-10 task, these images are passed into the model one at each time step, as a flattened $784 \times 1$ sequence.

Table 4: Average accuracy (6 runs or task orders) and standard deviation of different ACL models at the end of tasks on 20News dataset in class-IL and task-IL settings.

| | Class-IL | | | | | Task-IL | | | | |
|---|---|---|---|---|---|---|---|---|---|---|
| | Ceiling Methods | | ACL Methods | | | Ceiling Methods | | ACL Methods | | |
| | INDIV | MTL | FT | EWC | ER | INDIV | MTL | FT | EWC | ER |
| *20% Labelled Data* | | | | | | | | | | |
| RAND | 88.95 ±0.60 | 67.33 ±0.46 | 8.90 ±0.44 | 8.99 ±0.52 | 55.25†±2.34 | 88.95 ±0.60 | 89.27 ±0.52 | 63.34 ±2.48 | 63.23 ±1.93 | 87.81†±0.59 |
| ENT | 89.08 ±0.58 | 65.21 ±0.88 | 8.84 ±0.61 | 8.94 ±0.48 | 52.42†±1.42 | 89.08 ±0.58 | 89.10 ±0.46 | 61.41 ±4.81 | 61.38 ±7.03 | 88.47†±0.58 |
| MARG | **90.38** ±0.64 | **67.66** ±0.47 | 8.95 ±0.55 | **9.05** ±0.58 | 52.83†±2.24 | **90.38** ±0.64 | **89.80** ±0.38 | 62.41 ±2.63 | 61.18 ±2.38 | **88.65**†±0.23 |
| BADGE | 88.74 ±0.40 | 67.64 ±0.54 | 8.95 ±0.48 | 8.94 ±0.47 | **54.60**†±2.13 | 88.74 ±0.40 | 89.52 ±0.84 | 64.92 ±1.72 | **64.16** ±3.42 | 88.05†±0.54 |
| KMEANS | 88.67 ±0.74 | 67.41 ±0.68 | **9.49** ±1.33 | 8.88 ±0.50 | 54.54 ±2.38 | 88.67 ±0.74 | 89.23 ±0.45 | **68.32** ±1.98 | 64.04 ±3.75 | 87.47†±1.08 |
| *Full Labelled Data* | | | | | | | | | | |
| | 91.78 ±0.47 | 72.42 ±0.62 | 9.94 ±0.88 | 9.32 ±0.43 | 55.15†±1.00 | 91.78±0.47 | 91.17 ±0.69 | 70.48 ±3.25 | 77.45 ±1.56 | 89.33†±0.62 |

# D    MODEL TRAINING AND HYPER-PARAMETERS

We train the classifier using the Adam optimiser with a learning rate 1e-5, batch size of 16 sentences, up to 50 and 20 epochs for ASC and 20News respectively, with early stopping if there is no improvement for 3 epochs on the loss of the development set. All the classifiers are initialised with RoBERTA base. Each experiment is run on a single V100 GPU and takes 10-15 hours to finish. For experience replay, we use the fixed memory size of 400 for both ASC and 20News.

For the sequential MNIST datasets, we train an MLP as a classifier. Our learning rate is 0.01 and our batch size is 32. We used 10 epochs to train the model. The query size is 0.5% and our annotation budget is 10%. For sequential CIFAR10 dataset, the architecture of the model is Resnet18 with a learning rate of 0.05. Our batch size is 32. We use 30 epochs to train the model. The query size is 1% and the annotation budget is 25%.

# E    ADDITIONAL RESULTS AND ANALYSIS

## E.1    ACTIVE CONTINUAL LEARNING RESULTS ON TEXT CLASSIFICATION TASKS

**Domain Incremental Learning**    Table 5 reports the average accuracy at the end of all tasks on the ASC dataset in the domain-IL scenario. Overall, multi-task learning, CL and ACL models outperform individual learning (INDIV), suggesting positive knowledge transfer across domains. Compared to naive finetuning, EWC and ER tend to perform slightly better, but the difference is not significant. While using only 30% of the training dataset, ACL methods achieve comparable results with CL methods trained on the entire dataset (FULL). In some cases, the ACL with ER and EWC even surpasses the performance of the corresponding CL methods (FULL). In addition, the uncertainty-based AL strategies consistently outperform the diversity-based method (KMEANS).

**Class Incremental Learning**    The overall task performance on the 20News dataset is reported at Table 4. Contrasting to the domain-IL, the individual learning models (INDIV) surpass other methods by a significant margin in the class-IL scenario. The reason is that INDIV models only focus on distinguishing news from 2 topics while MTL, CL and ACL are 20-class text classification tasks. On the comparison of CL algorithms, finetuning and EWC perform poorly. This is inline with the previous finding that the regularisation-based CL method with pretrained language models suffers from the severe problem of catastrophic forgetting (Wu et al., 2022). While outperforming other CL methods significantly, ER still largely lags behind MTL. ACL performs comparably to CL (FULL) which is learned on the entire training dataset. The diversity-aware AL strategies (KMEANS and BADGE) outperform the uncertainty-based strategies (ENT and MARG). We speculate that ACL in non-overlap class IL is equivalent to the cold-start AL problem as the models are poorly calibrated, hence the uncertainty scores become unreliable.

**Task Incremental Learning**    Having task id as additional input significantly accelerates the accuracy of the CL methods. Both EWC and ER surpass the finetuning baseline. As observed in the class-IL, ER is the best overall CL method. Adding examples of previous tasks to the training

Table 5: Average accuracy (6 runs or task orders) and standard deviation of different ACL models at the end of tasks on ASC dataset (Domain-IL). The best AL score in each column is marked in **bold**. † denotes the best CL algorithm within the same AL strategy.

| | Ceiling Methods | | ACL Methods | | |
| | INDIV | MTL | FT | EWC | ER |
| --- | --- | --- | --- | --- | --- |
| *30% Labelled Data* | | | | | |
| RAND | 88.01 $\pm 1.78$ | 94.41 $\pm 0.80$ | 94.10$^\dagger$ $\pm 0.77$ | 93.03 $\pm 0.75$ | 93.71 $\pm 1.29$ |
| ENT | 91.51 $\pm 0.84$ | **94.95** $\pm 0.58$ | 95.08 $\pm 0.52$ | 95.58$^\dagger$ $\pm 0.33$ | 95.07 $\pm 0.55$ |
| MARG | **91.75** $\pm 0.74$ | 94.49 $\pm 0.83$ | **95.31** $\pm 0.55$ | **95.68**$^\dagger$ $\pm 0.56$ | **95.55** $\pm 0.38$ |
| BADGE | 83.61 $\pm 3.23$ | 93.85 $\pm 0.42$ | 92.68 $\pm 1.11$ | 93.15 $\pm 0.42$ | 94.24$^\dagger$ $\pm 0.49$ |
| KMEANS | 86.26 $\pm 2.42$ | 93.93 $\pm 0.60$ | 94.35$^\dagger$ $\pm 0.64$ | 93.59 $\pm 0.72$ | 93.76 $\pm 1.57$ |
| *Full Labelled Data* | | | | | |
| | 92.36 $\pm 0.84$ | 95.00 $\pm 0.27$ | 94.30 $\pm 1.09$ | 94.85$^\dagger$ $\pm 1.00$ | 94.19 $\pm 1.21$ |

Figure 11: Forgetting-Learning Profile of each ACL run on P-MNIST dataset (Domain-IL scenario).

minibatch of the current task resembles the effect of MTL in CL, resulting in a large boost in performance. Overall, ACL lags behind the FULL CL model, but the gap is smaller for ER.

### E.2 ACTIVE CONTINUAL LEARNING RESULTS ON IMAGE CLASSIFICATION TASKS

**Results on Permuted MNIST** We plot the forgetting-learning profile of P-MNIST in Figure 11. In general, ER has lower forgetting rate than FT and EWC. However, it has a slightly slower learning rate than EWC.

### E.3 INDEPENDENT VS. SEQUENTIAL LABELLING

We show the difference of the performance between CL with independent AL and the corresponding ACL method for P-MNIST, S-MNIST and S-CIFAR10 in Table 7, Table 8 and Table 9 respectively.

### E.4 ADDITIONAL ANALYSIS

**Task Prediction Probability** Figure 12 shows the task prediction probability at the end of the training for ACL with ER and entropy methods in class-IL scenario. The results of other methods can be found in the appendix. Overall, the finetuning and EWC methods put all the predicted probability on the final task, explaining their inferior performance. In contrast, ER puts a relatively small probability on the previous tasks. As expected, the sequential labelling method has a higher task prediction probability on the final task than the independent labelling method, suggesting higher forgetting and overfitting to the final task.

**AL Query Overlap** We report the Jaccard coefficient between the selection set of sequential and independent entropy-based AL methods at the end of AL for each task in the task sequence in Figure 13. We observe low overlapping in the uncertainty-based AL methods. On the other hand, diversity-based AL methods have high overlap and decrease rapidly throughout the learning of subsequent tasks.

Table 6: Average accuracy (6 runs or task orders) and standard deviation of different ACL models at the end of tasks on image classification benchmarks.

| | | Ceiling Methods | | ACL Methods | | | | | | | |
|---|---|---|---|---|---|---|---|---|---|---|---|
| | | INDIV | MTL | FT | EWC | ER | iCaRL | GDUMB | DER | DER++ | AGEM |
| **S-MNIST Class-IL** | | | | | | *10% Labelled Data* | | | | | |
| | RAND | 99.73±0.03 | 93.43±0.20 | 19.97±0.02 | 20.01±0.08 | **87.81**±0.66 | **77.88**±2.05 | 85.34±0.82 | 78.51±3.11 | 78.80±3.29 | 50.60±9.26 |
| | ENT | 99.74±0.03 | 95.76±0.19 | 19.97±0.02 | 17.46±3.07 | 79.38±3.00 | 73.59±3.80 | 76.06±2.74 | 67.19±5.50 | 67.19±5.50 | 39.40±7.19 |
| | MARG | 99.74±0.03 | 96.32±0.16 | **19.98**±0.01 | 19.72±0.47 | 79.89±1.98 | 75.66±2.85 | 80.54±1.78 | 65.69±8.41 | 64.48±7.62 | 39.91±8.17 |
| | BADGE | 99.74±0.03 | 93.20±0.33 | 19.92±0.06 | 20.08±0.13 | 87.40±0.44 | 76.35±2.36 | **85.35**±0.61 | **83.25**±3.31 | 78.48±5.59 | 56.52±3.80 |
| | CORESET | **99.75**±0.03 | 95.26±0.24 | **19.98**±0.01 | 21.27±1.91 | 83.12±1.16 | 73.51±5.98 | 78.80±2.17 | 74.17±6.21 | 74.17±6.21 | 38.48±7.19 |
| | | | | | | *Full Labelled Data* | | | | | |
| | FULL | 99.71±0.02 | **97.57**±0.15 | **19.98**±0.01 | 20.41±0.71 | 84.68±1.88 | 76.69±0.80 | 84.52±1.49 | 73.77±8.63 | **88.64**±1.58 | **56.89**±9.10 |
| **S-MNIST Task-IL** | | | | | | *10% Labelled Data* | | | | | |
| | RAND | 99.73±0.03 | 99.07±0.03 | **78.42**±3.33 | 94.34±3.32 | 98.75±0.30 | 97.54±0.25 | **97.48**±0.08 | 98.30±0.57 | 96.96±0.64 | 97.92±0.29 |
| | ENT | 99.74±0.03 | 99.69±0.04 | 75.31±5.33 | **95.81**±3.54 | 98.78±0.39 | **98.08**±0.88 | 97.23±0.22 | 98.54±0.47 | 95.85±1.42 | 97.38±1.02 |
| | MARG | 99.74±0.03 | 99.62±0.04 | 74.81±5.24 | **95.81**±3.54 | 98.70±0.42 | 96.71±1.09 | 97.23±0.22 | 98.54±0.47 | 96.25±0.77 | 96.85±0.60 |
| | BADGE | 99.74±0.03 | 98.93±0.10 | 74.81±5.24 | **95.81**±3.54 | **98.87**±0.03 | 97.02±0.54 | 97.14±0.31 | 98.43±0.23 | 96.85±1.95 | 97.98±0.44 |
| | CORESET | **99.75**±0.03 | 99.54±0.05 | 78.39±6.42 | 94.11±4.44 | 98.83±0.12 | 97.58±2.44 | 97.09±0.32 | 98.39±0.36 | 97.48±0.53 | 98.07±0.60 |
| | | | | | | *Full Labelled Data* | | | | | |
| | FULL | 99.71±0.02 | 99.71±0.04 | 77.82±7.09 | 94.38±3.81 | 98.79±0.10 | 97.22±5.38 | 97.03±0.64 | **98.76**±0.12 | **98.73**±0.16 | **98.65**±0.18 |
| **S-CIFAR10 Class-IL** | | | | | | *25% Labelled Data* | | | | | |
| | RAND | 86.24±0.45 | 66.40±1.34 | 17.11±1.10 | 16.95±0.74 | 24.12±2.20 | 35.21±2.29 | 26.68±4.24 | 17.67±0.62 | 18.32±0.77 | 17.36±1.12 |
| | ENT | 87.16±0.55 | 65.08±3.04 | 17.46±1.01 | 17.17±1.03 | **25.20**±3.79 | 35.74±1.75 | 22.89±2.40 | 17.90±1.18 | 17.71±1.39 | 17.88±0.95 |
| | MARG | 87.47±0.35 | 68.66±1.10 | 17.55±1.05 | 17.08±0.90 | 24.28±4.25 | 34.02±3.82 | 24.32±2.71 | 17.76±1.15 | 17.34±1.08 | 17.93±0.90 |
| | BADGE | 84.99±0.23 | 66.27±1.19 | 17.40±1.01 | 16.95±0.98 | 24.35±1.83 | 31.75±3.22 | **29.26**±1.93 | 17.51±1.07 | 17.27±1.11 | 17.48±0.93 |
| | CORESET | 86.39±0.41 | 67.33±1.48 | 17.26±1.23 | 17.04±0.85 | 23.58±2.57 | 35.39±2.79 | 22.03±4.78 | 17.57±0.70 | 17.45±0.92 | 17.72±0.89 |
| | | | | | | *Full Labelled Data* | | | | | |
| | FULL | **94.99**±0.36 | **89.60**±0.73 | **19.09**±0.42 | **19.20**±0.39 | 23.26±4.21 | **43.70**±3.66 | 27.57±3.12 | **19.16**±0.35 | **28.85**±6.39 | **21.32**±1.86 |
| **S-CIFAR10 Task-IL** | | | | | | *25% Labelled Data* | | | | | |
| | RAND | 86.24±0.45 | 90.75±0.81 | 59.65±3.50 | 69.03±2.51 | 76.39±1.11 | 77.07±2.18 | 67.23±1.69 | 70.55±2.00 | 69.62±3.54 | 71.34±2.46 |
| | ENT | 87.16±0.55 | 90.18±1.07 | **59.93**±5.09 | 67.80±2.36 | 73.96±1.93 | 77.93±2.27 | 63.23±3.05 | 69.45±3.55 | 68.65±3.64 | 66.67±2.59 |
| | MARG | 87.47±0.35 | 91.20±0.83 | 58.18±3.42 | **69.50**±2.62 | 75.65±1.86 | 78.93±1.35 | 67.19±2.37 | 63.92±1.70 | 65.34±4.89 | 63.58±4.42 |
| | BADGE | 84.99±0.23 | 90.84±0.46 | 55.52±1.77 | 68.39±3.97 | 75.96±1.92 | 78.93±3.23 | 67.36±2.21 | 67.89±4.20 | 71.93±2.71 | 67.51±1.94 |
| | CORESET | 86.39±0.41 | 91.13±0.39 | 58.08±2.70 | 68.75±3.98 | 72.86±2.53 | **79.32**±3.36 | 65.21±2.80 | 66.18±1.10 | 70.26±5.20 | 63.98±2.30 |
| | | | | | | *Full Labelled Data* | | | | | |
| | FULL | **94.99**±0.36 | **97.86**±0.18 | 59.55±2.22 | 65.00±5.14 | **83.47**±1.72 | 77.76±4.16 | **70.23**±2.09 | **77.16**±4.14 | **86.25**±0.75 | **80.90**±6.00 |
| **P-MNIST DOMAIN-IL** | | | | | | *5% Labelled Data* | | | | | |
| | RAND | 94.33±0.12 | 92.49±0.36 | 55.83±2.81 | 38.55±2.27 | **75.10**±0.75 | | 53.61±0.82 | 86.22±0.31 | 86.23±0.39 | **74.09**±1.10 |
| | ENT | 97.04±0.04 | 95.00±0.10 | 44.72±2.06 | 37.83±1.07 | 65.08±1.16 | | 42.28±1.59 | 76.02±1.15 | 75.72±1.07 | 63.89±0.91 |
| | MARG | 97.23±0.03 | 95.38±0.14 | 44.07±3.03 | 38.15±1.26 | 65.82±1.49 | | 48.52±1.88 | 77.91±0.69 | 77.82±0.63 | 64.42±0.70 |
| | BADGE | 93.80±0.07 | 92.19±0.38 | **58.44**±1.07 | 39.02±1.93 | 74.99±1.35 | | **54.04**±1.30 | **86.64**±0.36 | **86.62**±0.46 | 74.00±0.94 |
| | CORESET | 96.24±0.04 | 94.11±0.17 | 46.55±1.93 | 35.35±2.06 | 67.90±1.00 | | 42.79±1.95 | 77.60±1.39 | 76.94±0.83 | 67.20±0.73 |
| | | | | | | *Full Labelled Data* | | | | | |
| | FULL | **97.62**±0.05 | **96.69**±0.14 | 35.12±2.31 | **40.13**±2.30 | 60.13±0.65 | | 53.18±0.98 | 71.49±1.62 | 72.63±0.85 | 58.51±3.22 |

Table 7: Relative performance of CL with independent AL with respect to the corresponding ACL for P-MNIST.

| | FT | EWC | ER | GDumb | DER | DER++ | AGEM |
|---|---|---|---|---|---|---|---|
| IND. RAND | +3.56 | +0.09 | −0.04 | +0.48 | +0.54 | +0.81 | −0.67 |
| IND. ENT | +0.35 | −4.46 | −2.32 | −9.08 | −5.59 | −4.08 | −2.36 |
| IND. MARG | +3.82 | −3.49 | −0.72 | −9.97 | −3.84 | −3.99 | −2.24 |
| IND. BADGE | −6.30 | −0.82 | +0.03 | +1.65 | −0.99 | −0.08 | −1.29 |

Table 8: Relative performance of CL with independent AL with respect to the corresponding ACL for S-MNIST.

| | S-MNIST (Class-IL) | | | | | | | | S-MNIST (Task-IL) | | | | | | | |
|---|---|---|---|---|---|---|---|---|---|---|---|---|---|---|---|---|
| | FT | EWC | ER | iCaRL | GDumb | DER | DER++ | AGEM | FT | EWC | ER | iCaRL | GDumb | DER | DER++ | AGEM |
| IND. RAND | −0.10 | +3.60 | +0.70 | −1.39 | +0.42 | +4.20 | +3.90 | +2.18 | +3.82 | −1.33 | −0.12 | −0.15 | −0.10 | −1.83 | −0.43 | +0.08 |
| IND. ENT | −0.30 | +3.59 | +1.83 | −3.30 | −3.72 | −5.06 | −5.29 | +1.31 | −0.35 | −10.70 | −0.63 | −0.38 | −3.65 | −5.45 | −2.76 | −1.30 |
| IND. MARG | −0.19 | −1.94 | +0.06 | −6.38 | −7.43 | −4.93 | −6.46 | +2.37 | +1.68 | −11.91 | −0.52 | +0.53 | −3.53 | −5.66 | −3.38 | +0.89 |
| IND. BADGE | −0.06 | +6.14 | +0.59 | +0.32 | +0.25 | +0.15 | +4.92 | +2.59 | +3.35 | −0.64 | −0.47 | −0.30 | −0.07 | −0.78 | +0.83 | −0.05 |
| IND. CORESET | −0.07 | +0.66 | −0.60 | +1.77 | −0.22 | −3.98 | −3.31 | +2.28 | −6.58 | −4.94 | −0.57 | +0.29 | −2.18 | −3.49 | −2.58 | −0.79 |

Table 9: Relative performance of CL with independent AL with respect to the corresponding ACL for S-CIFAR10.

| | S-CIFAR10 (Class-IL) | | | | | | | | S-CIFAR10 (Task-IL) | | | | | | | |
| --- | --- | --- | --- | --- | --- | --- | --- | --- | --- | --- | --- | --- | --- | --- | --- | --- |
| | FT | EWC | ER | iCaRL | GDumb | DER | DER++ | AGEM | FT | EWC | ER | iCaRL | GDumb | DER | DER++ | AGEM |
| IND. RAND | +0.05 | −0.43 | +0.53 | −1.36 | +0.66 | −0.66 | −1.24 | −0.15 | −3.63 | +0.18 | −0.61 | +0.96 | +1.01 | −1.43 | +0.52 | −0.96 |
| IND. ENT | −0.19 | −0.41 | −1.47 | −6.85 | −2.55 | −0.70 | −0.75 | −0.35 | −4.97 | +1.35 | −2.88 | +0.95 | −3.35 | −2.99 | −0.31 | −0.32 |
| IND. MARG | −0.01 | −0.76 | −1.82 | −2.19 | −1.44 | −0.22 | +0.08 | −0.59 | −1.71 | −1.87 | −3.85 | +1.60 | −9.02 | +5.63 | +0.03 | −0.25 |
| IND. BADGE | −0.48 | −0.48 | −2.10 | −0.18 | −2.82 | −0.83 | −0.55 | −0.63 | +0.75 | −4.43 | +1.29 | −2.36 | +0.75 | +3.80 | −3.02 | +2.34 |
| IND. CORESET | +0.07 | −0.25 | −0.62 | −5.12 | +2.67 | −0.55 | −0.07 | −0.38 | −0.62 | −0.18 | +1.85 | −0.67 | −1.96 | +3.52 | −0.65 | +3.16 |

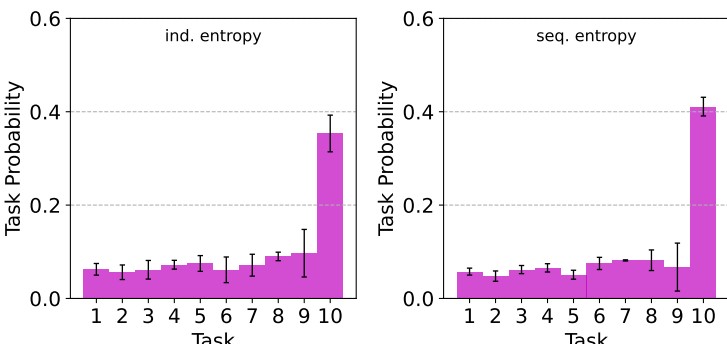

Figure 12: Task prediction probability of ER-based ACL on 20News dataset (Class-IL).

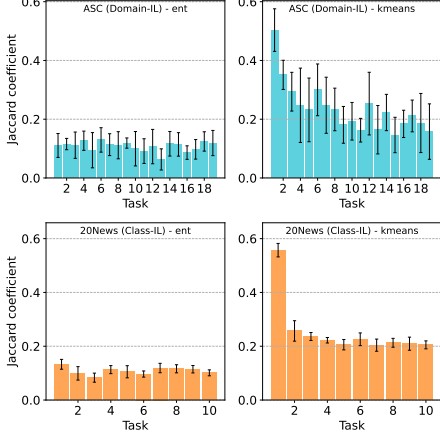

Figure 13: Jaccard coefficient between independent and sequential labelling queries of ER-based ACL methods.

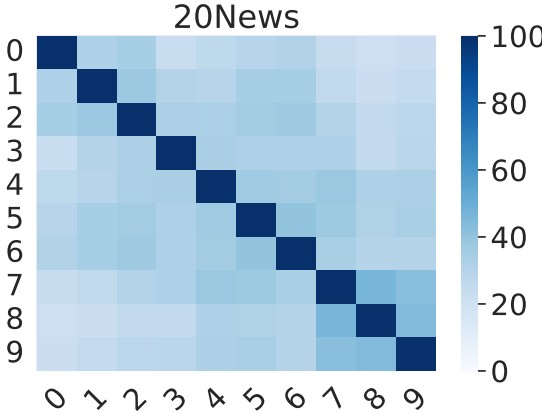

Figure 14: Task similarity in 20News dataset.

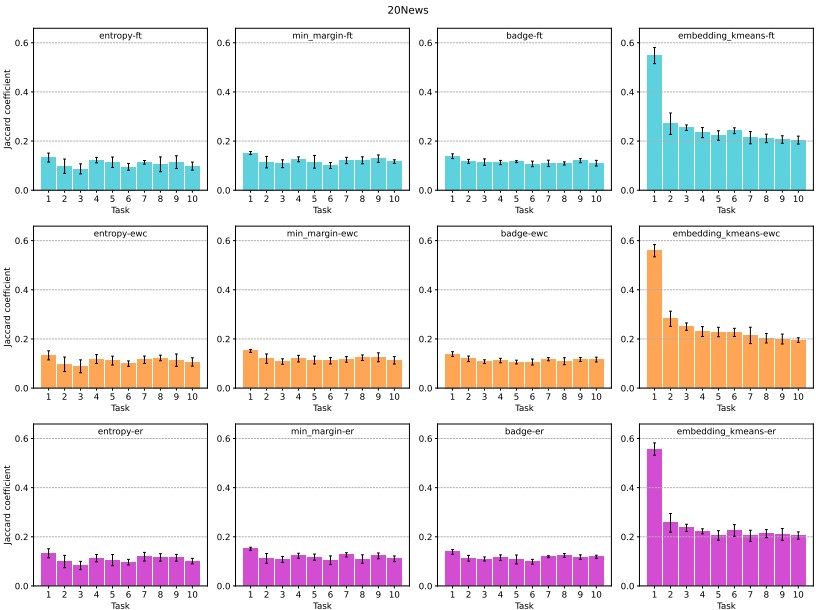

Figure 15: Jaccard coefficient between sequential and independent AL queries of ACL methods on 20News (class-IL).

**Task similarity.** We examine the task similarity in 20News dataset by measuring the percentage of vocabulary overlap (excluding stopwords) among tasks. It can be shown in Figure 14 that tasks in 20News dataset have relatively low vocabulary overlap with each other (less than 30%). This explains the poor performance of EWC as it does not work well with abrupt distribution change between tasks (Ke et al., 2021).

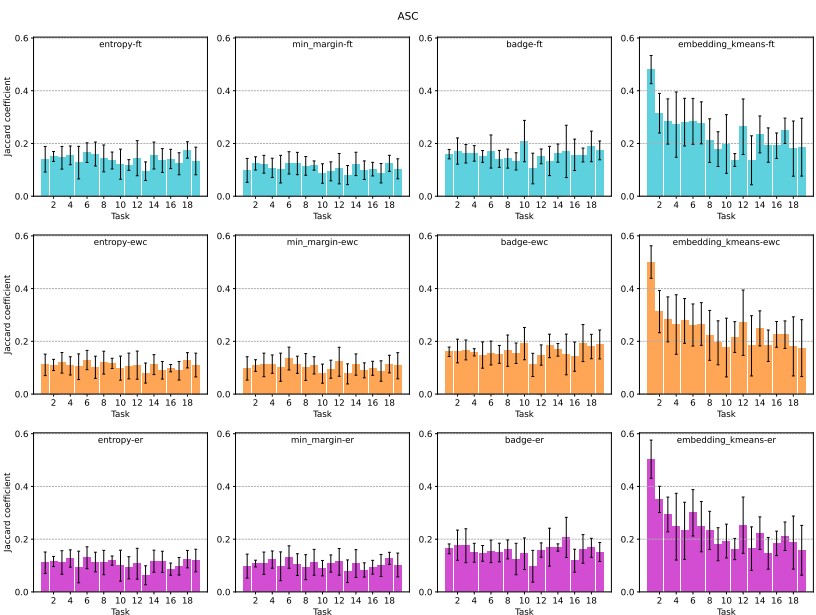

Figure 16: Jaccard coefficient between sequential and independent AL queries of ACL methods on ASC.

