# OpenReview forum: "Active Continual Learning: On Balancing Knowledge Retention and Learnability"
_ICLR.cc/2024/Conference — Submitted to ICLR 2024_

### Official Review · Reviewer_Ej6r · 2023-10-29

**Soundness:** 2 fair
**Presentation:** 2 fair
**Contribution:** 2 fair
**Rating:** 6
**Confidence:** 4

**Summary:**

This paper explores the domain of active continual learning, a novel problem that, to the best of my knowledge, has not been previously investigated. It doesn't aim to introduce a new model; instead, it serves as an analytical study, delving into the interplay between active learning and continual learning.

**Strengths:**

1. The problem is novel and interesting
2. The analysis experiments are extensive and should be valuable to the community

**Weaknesses:**

At first glance, this paper encompasses both text and image datasets, along with three CL settings. However, upon closer examination of the baseline, it appears that the baselines are largely drawn from Class-Incremental Learning (CIL), such as DER++, iCaRL, and GDumb. Consequently, the paper doesn't utilize any state-of-the-art methods for specific settings and data types. For example, in the Text-Incremental Learning (TIL) for text data, [1,2] represent more likely SoTa approaches (further details available in the survey [3]). This lack of a fair comparison with state-of-the-art methods is the major drawback for this analysis paper.

[1]: Adapting BERT for Continual Learning of a Sequence of Aspect Sentiment Classification Tasks, Ke et al., 2021
[2]: https://arxiv.org/abs/2310.09436
[3]: https://arxiv.org/abs/2211.12701

**Questions:**

See above

---

> ### Author Response · Authors · 2023-11-16
>
> We thank the reviewer for their useful feedback. Please see our responses addressing the specific concerns below
>
> > **it appears that the baselines are largely drawn from Class-Incremental Learning (CIL), such as DER++, iCaRL, and GDumb.**
> > **The paper doesn't utilize any state-of-the-art methods for specific settings and data types.**
>
> As stated in the limitation section, we only consider several well-studied algorithms and classification problems. The rationale behind this is twofold. The straightforward nature of classification tasks and the well-established algorithms facilitate identifying factors that influence the performance and behaviour of ACL. To provide a holistic view, we carefully select the representative AL and CL methods to cover: (i) most non-architectural CL families: regularization (EWC) and rehearsal (ER); and (ii) AL families: uncertainty-based (entropy, min-margin), diversity-based (embedding k-means) and combination of uncertainty and diversity (BADGE).
>
> This deliberate decision aligns with our intention to establish a strong foundational understanding of the principles underpinning ACL. By beginning with well-established and relatively simpler algorithms, we set a solid baseline for understanding the behaviour of ACL before tackling more complex algorithms. However, we acknowledge the potential for broader exploration and leave them as future works.
>
> In considering these responses to your feedback, we hope you will consider increasing your score for our paper.

---

### Official Review · Reviewer_j1Rg · 2023-10-31

**Soundness:** 2 fair
**Presentation:** 2 fair
**Contribution:** 2 fair
**Rating:** 3
**Confidence:** 2

**Summary:**

This paper presents an experimental analysis exploring the potential of active learning for annotation of examples for continual learning. The paper considers a variety of continual learning settings, including domain-incremental, class-incremental and task-incremental learning, and examines aspects such as the balance between forgetting and learning in CL aided by active learning. The authors conduct experiments in six benchmark datasets + tasks, including P-MNIST, MNIST and CIFAR-10, and over a range of state-of-the-art continual learning methods for regularization and example replay. In the experiments, the authors examine the performance of these methods integrated with the ACL proposed approach, and compare this to joint learning and multitask learning with respect to overall accuracy, forgetting, and learning-forgetting profile.

**Strengths:**

- The paper examines a reasonable avenue for continual learning, which is selecting annotated examples by means of active learning. The paper aims at answering three important research questions in this setting. This demonstrates the originality of the paper.
- The paper is in general well organized and the concepts are presented clearly and to sufficient depth.

**Weaknesses:**

- The choice of some visualizations in the paper is very odd. For example, in Figures 3 and 4, a dashed red line is selected to represent the performance on the "full labelled dataset". But, why is a line used to connect in between methods (x axis)? What is the meaning of this? Similarly, all the dots representing different strategies make it very difficult to grasp what is the actual performance of each of the selected methods with each of those strategies. I would strongly suggest to find a much better representation.
- Although the selected datasets are CL benchmarks, these are also the easiest ones. I would have expected to see experimental results on more challenging datasets such as CIFAR-100, some version of ImageNet (tiny-ImageNet, mini-ImageNet), etc. Furthermore, from Figure 3 and 4, it seems that for the slightly more challenging datasets and tasks (P-MNIST, CIFAR-10 CIL, TIL), all the ACL methods perform substantially badly, therefore raising questions on the actual effectiveness of CL combined with active learning, and the significance of the proposed approach.
- A final remarkable weakness that I see in this paper is the limited number of tasks in the experiments. I would expect that active selection of examples would be significantly more difficult as the number of tasks increases, and therefore would have expected to see results along these lines.

**Questions:**

- Please refer to questions in the "weaknesses" section.

---

> ### Author Response · Authors · 2023-11-16
>
> We thank the reviewer for their useful feedback. Please see our responses addressing the specific concerns below
>
> > **The choice of some visualizations in the paper is very odd. For example, in Figures 3 and 4, a dashed red line is selected to represent the performance on the "full labelled dataset". But, why is a line used to connect in between methods (x axis)? What is the meaning of this?**
>
> Thanks for the comment. We aimed to establish the baseline to compare between CL with full labelled dataset and CL with AL - active continual learning (ACL). We re-plot Figure 3 and 4 to report the relative performance of ACL and the baseline. We hope they are better visualised.
>
> > **Furthermore, from Figure 3 and 4, it seems that for the slightly more challenging datasets and tasks (P-MNIST, CIFAR-10 CIL, TIL), all the ACL methods perform substantially badly, therefore raising questions on the actual effectiveness of CL combined with active learning, and the significance of the proposed approach.**
>
> We would like to emphasize that our goal is not to show that ACL is better than CL. In fact, we propose a practical scenario where there is a need to label data for CL. Our work empirically show that simply combining existing CL and AL is not optimal due to the trade-off between forgetting and learnability. We hope that our research will give rise to new avenues of research, prompting the community to further research regarding the development of ACL methods capable of effectively harmonizing the dual objectives of low forgetting and quick learnability.
>
> > **A final remarkable weakness that I see in this paper is the limited number of tasks in the experiments. I would expect that active selection of examples would be significantly more difficult as the number of tasks increases, and therefore would have expected to see results along these lines.**
>
> We evaluate on both text classification and image classification tasks which cover different CL scenarios (domain-IL, class-IL and task-IL) and having different number of tasks: MNIST (5 tasks), CIFAR-10 (5 tasks), ASC (19 tasks) and 20News (10 tasks).
>
> In considering these responses to your feedback, we hope you will consider increasing your score for our paper.

---

### Official Review · Reviewer_Hffh · 2023-11-01

**Soundness:** 3 good
**Presentation:** 3 good
**Contribution:** 2 fair
**Rating:** 5
**Confidence:** 3

**Summary:**

The paper has full study for active continual learning problem that explores the tradeoff between a quick learner and not to forget the learned knowledge.

**Strengths:**

The AL + CL have broad interest. There are many evaluations of current AL and CL approaches in paper.

**Weaknesses:**

The paper has the following concerns that I am afraid it cannot meet the ICLR threshold.

1) The paper does not seem to have a high novel approach, but mostly evaluate the current AL and CL approach.

2) The evaluation seems only on P-MNIST, S-MNIST, CIFAR-10, maybe small sets.

**Questions:**

What do you think would be the good application of ACL?

---

> ### Author Response · Authors · 2023-11-16
>
> We thank the reviewer for their useful feedback. Please see our responses addressing the specific concerns below
>
>
> > **The paper does not seem to have a high novel approach, but mostly evaluate the current AL and CL approach.**
>
> We would like to emphasize that our contributions extend well beyond introducing a formulation for ACL:
> - We introduce the analysis of the forgetting-learning profile to study the trade-off between low forgetting and quick learnability objectives in the context of ACL.
> - While this trade-off has been previously studied in the CL literature, our empirical findings confirm the existence of this trade-off in the simple combination of the AL and CL methods.
> - We conduct comprehensive experiments on combining several representative CL and AL methods in domain, class, and task incremental learning. The findings provide insights into the benefit of ACL (sequential vs independent labelling) and the selection of AL and CL methods in each CL learning scenario.
>
> Our findings give rise to new avenues of research, prompting the community to further research regarding the development of ACL methods capable of effectively harmonizing the dual objectives of low forgetting and quick learnability.
>
>
> > **The evaluation seems only on P-MNIST, S-MNIST, CIFAR-10, maybe small sets.**
>
> We evaluation on both text classification and image classification tasks which covering different CL scenarios (domain-IL, class-IL and task-IL) and having different number of tasks: MNIST (5 tasks), CIFAR-10 (5 tasks), ASC (19 tasks) and 20News (10 tasks).
>
>
> > **What do you think would be the good application of ACL?**
>
> One motivation of CL is that all the training data are not available at once. Instead, we often have the situation to adapt or integrate new knowledge into the model with the arrival of new training data. In this paper, we pay attention to the training data preparation, i.e. labelling data for new tasks. We study whether it is beneficial to take the previous tasks and current model into account when labelling new training data.
>
> For instance, let's consider the fraud detection system. As time progresses, novel types of fraudulent activities might emerge. Consequently, there arises a necessity to periodically update the detection system in the production. The new training batch can be labelled independently or sequentially via active learning. We hypothesize that active learning can assist in selecting data points that offer the most gain to the existing systems.
>
> In considering these responses to your feedback, we hope you will consider increasing your score for our paper.

---

### Official Review · Reviewer_3Lkg · 2023-11-02

**Soundness:** 2 fair
**Presentation:** 2 fair
**Contribution:** 2 fair
**Rating:** 5
**Confidence:** 4

**Summary:**

The authors address the task of active continual learning, in which a sequence of tasks consisting of unlabeled data and an annotation budget are presented to the model. This is challenging because of the need of balancing the ability to not forgetting and quickly learning within the annotation budget. They propose a forgetting-learning profile to better understand the behavior of active continual learners and guidelines to choose active learning and continual learning algorithms. Specifically, they show that uncertainty-based active learning is better suited for domain incremental learning, whereas diversity-based active learning is better for task incremental learning.

**Strengths:**

There is a need for a systematic evaluation of continual active learning models given the variety of options for each of the main components of the system, namely, active learning and continual learning.

The authors took the time and effort to answer to the criticism raised by the reviewer in the weaknesses and question sections below, which increased the score relative to the initial value.

**Weaknesses:**

Presenting ACL results without error bars (consistently) has the potential of misrepresenting the capabilities of different approaches and a missed opportunity to also highlight the stability/consistency of different approaches.

The analysis of the results in Figure 3 (and Figure 4) is weak, for instance "ACL achieves comparable performance to CL in most scenarios", however, the statement seems to be purely qualitative. Why is it expected that experience replay methods achieve high accuracy than FT and EWC. Also, the Figure is difficult to read. Alternatively, showing accuracy relative to full will make for a better scaling and visualization of the differences between AL approaches.

Figures 8 and 9 are very difficult to read with so many dots with similar colors and similar to 3 and 4, the analysis is somewhat shallow given the amount of data in the Figures and what is not shown (due to error bars).

For a purely experimental contribution (with no methodological novelty) the design of the experiment lacks a systematic evaluation of the factors affecting the performance of CL and Al methods. For instance, number of tasks, backbone, model size, annotation budget (especially the annotation budget), computational budget, etc. Moreover, the analysis of the results is mostly qualitative instead of a quantitative evaluation of the significance of the improvements of different methods.

**Questions:**

How was Figure 1(b) obtained?

Why not showing Figures 5 and 6 like in Figure 1(b) to better show the trade-off between forgetting rate and LCA?

Figures 5, 6 and 7 have error bars, however, (in the paper proper) it is not described how they were obtained.

---

> ### Author Response · Authors · 2023-11-16
>
> Thank you for your detailed feedbacks. Following your suggestion, we have improved the visualization of Figure 3-6 in our revised manuscript. Please see our responses addressing the specific concerns below
>
> > **ACL results without error bars (consistently) has the potential of misrepresenting the capabilities of different approaches and a missed opportunity to also highlight the stability/consistency of different approaches**
>
> We add the error bar to Figure 3,4 and 8-9. The average accuracy with standard deviation of all datasets are reported in Table 4-6 in the Appendix.
>
> >  **showing accuracy relative to full will make for a better scaling and visualization of the differences between AL approaches**
>
> Thanks for the suggestion. We re-plot figure 3 and 4 to highlight the differences with respect to CL on the full labelled data in the revised version.
>
> > **Why is it expected that experience replay methods achieve high accuracy than FT and EWC?**
>
> Experience replay methods have accessed to a small amount of training data from previous tasks, hence it is less prone to catastrophic forgetting and have better accuracy on previous task. We updated our manuscript to make the explanation clear.
>
> > **Figures 8 and 9 are very difficult to read with so many dots with similar colors and similar to 3 and 4, the analysis is somewhat shallow given the amount of data in the Figures and what is not shown (due to error bars)**
>
> The aim of figure 8 and 9 is to provide an illustration on the trade-off between forgetting and learnability. It shows how current ACL methods reside in distinct regions in the forgetting and learnability space. We will add the error bars into these figures.
>
> >  **the design of the experiment lacks a systematic evaluation of the factors affecting the performance of CL and Al methods. For instance, number of tasks, backbone, model size, annotation budget (especially the annotation budget), computational budget, etc**
>
> Thanks for the suggestion. Our experiment design focus on the combination of AL and CL in different CL scenario (domain-IL, class-IL and task-IL) and different modalities (image and text). The evaluated tasks are varied in the number of tasks: MNIST (5 tasks), CIFAR-10 (5 tasks), ASC (19 tasks) and 20News (10 tasks).
> The annotation budget for each task is chosen based on the AL curve of the individual task such that the performance does not increase much as new labeled data arrives. We have an analysis of the forgetting rate of ACL at different AL budgets.
>
> We acknowledge that it would be interesting to explore other factors, such as backbone, model size on the performance of ACL. However, due to the limitation of computational resources, we leave this exploration as future works.
>
> > **How was Figure 1(b) obtained?**
>
> Firgure 1(b) is an illustration based on our observations in Figure 8 and 9.
>
> > **Why not showing Figures 5 and 6 like in Figure 1(b) to better show the trade-off between forgetting rate and LCA?**
>
> Thanks for the suggestion. We make the change accordingly.
>
> > **Figures 5, 6 and 7 have error bars, however, (in the paper proper) it is not described how they were obtained.**
>
> We added a description of how they were obtained in the revised version.
>
> > **The analysis of the results is mostly qualitative instead of a quantitative evaluation of the significance of the improvements of different methods.**
>
> We updated the visualization and added a detailed accuracy and standard deviation in the manuscript. We believe that it would highlight better quantitative results of our study.
>
> In considering these responses to your feedback, we hope you will consider increasing your score for our paper.

---

> > ### Comment · Reviewer_3Lkg · 2023-11-22
> >
> > Thanks to the authors for the amount of work and thought put into their rebuttal. The score has been updated accordingly.

---

> ### Author Response · Authors · 2023-11-22
>
> We appreciate your decision to improve the score. Thanks for your consideration and time to read through the rebuttal.

---

### Author Response · Authors · 2023-11-16

We would like to thank all reviewers for the insightful reviews and comments.

One concern among the reviewers is the ambiguity in our visualization. We took this into account and re-ploted Figure 3 and 4 to better highlight the performance between continual learning baseline (CL) and active continual learning (ACL). The changes are highlighted in red in our revised manuscript. We also added error bar into Figure 3-4 and the learning-forgetting profile in figure 8-9.

---

### Meta-Review · Area_Chair_csqa · 2023-12-05

**Metareview:**

The paper examines potential of combining continual learning and active learning. It considers a range of settings, such as domain-incremental, class-incremental and task-incremental. The evaluation is done on six benchmark dataset, covering vision and language. The reviewers overall appreciate well-organized and extensive (Ej6r) experiments (reviewer j1Rg), and think this could be valuable for the community.  The reviewers have concerns with presentation (including visualization, showing statistical significance of results). There were also concerns with only considering relatively *easy* datasets, while there is plethora of more challenging benchmarks. Also, they find the baselines to be relatively week (Reviewer Ej6r). Overall, despite reviewers acknowledging the value of this research direction, this paper will likely benefit from another round of reviews, accommodating changes suggested by reviewers.

**Justification For Why Not Higher Score:**

The choice of evaluation datasets.

**Justification For Why Not Lower Score:**

The experiments are solid and the research question is well motivated.

---

### Decision · Program_Chairs · 2024-01-16

Reject